# The influence of culture on open defaecation in some basic schools in selected districts in Ghana: A preliminary study

Saviour Victor Adjibolosoo[1], Benjamin Dankyira Ofori[1], Philip Baba Adongo[2], Stephen Afranie[3], Dzidzo Regina Yirenya-Tawiah[1]*

1 Institute for Environment and Sanitation Studies (IESS), College of Basic and Applied Sciences, University of Ghana, Legon, Ghana, 2 School of Public Health, College of Health Sciences, University of Ghana, Legon, Ghana, 3 Department of Social Work, College of Humanities, University of Ghana, Legon, Ghana

* dzidzoy@staff.ug.edu.gh

**Data Availability Statement:** All relevant data are within the manuscript and its Supporting Information files.

## Abstract

Open defaecation has remained a major public and environmental health concern which has gained global attention. This study explored cultural factors including superstition, taboo, norms and values influencing open defaecation behaviour among basic school pupils in the Eastern and Volta Regions all within Southern Ghana. Self-reported data were collected from 400 pupils using structured questionnaire and focus group discussions. Descriptive, bivariate and multivariate statistics were used to analyze the quantitative data. The qualitative data was analyzed using thematic content analytical procedure. The results of the study showed, superstitions, traditional norms and taboo yielded statistically significant effect sizes with pupils' open defaecation behaviour: and superstition: $r = 0.728$, $p < 0.05$; traditional norms: $r = 0.425$, $p < 0.05$; taboos: $r = 0.462$, $p < 0.05$. The study concluded that superstitions and traditional norm and taboo constituted the key cultural drivers influencing pupils' open defaecation behaviour in the Eastern and Volta Regions. It is recommended that the Ministry of Education should incorporate open defaecation issues into the educational curriculum and develop culturally sensitive educational programs for a massive educational campaign to stop open defaecation in the schools. To ensure that the messaging for these campaigns resonate with target audience, communication campaigns should promote a number of positive emotional and social issues related to improved social status and positive self-esteem, better growth and economic opportunities with toilet use.

## Introduction

Sanitation is a major cross cutting issue that links 12 out of the 17 Sustainable Development Goals (SDG) giving it prominence on the global agenda. Worldwide, 4.2 billion people live without safely managed sanitation and 673 million still practice open defecation [1]. Open defaecation creates a conducive environment for pathogens that thrive in faecal matter and cause diseases such as diarrhoea, typhoid, schistosomiasis and many other neglected tropical

**Funding:** Research was funded through Carnegie Corporation of New York on the project "Next Generation of Academics in Africa: Enhancing the University of Ghana's Capacity to Deliver Post-graduate Research and Training" University of Ghana (ECBAS 035/15-16) to SVA.

**Competing interests:** The authors have declared that no competing interests exist.

diseases to spread. Children, particularly those under five years, suffer high mortality from some of these diseases such as diarrhoea while others suffering from schistosomiasis for example, may be left stunted or suffer cognitive disabilities [2, 3]. It is estimated that 432,000 diarrhoea deaths occur annually in the world [4].

Previous reports had shown inadequate water and sanitation facilities in schools as major hindrances towards achievement of the Millennium Development Goals (MDGs) a situation propelling sanitation to gain focus on the SDG agenda. Unfortunately, sanitation coverage in schools continues to remain an issue with the lowest coverage in sub Saharan Africa [5]. Globally, 23% of schools have no sanitation service at all and fewer than 50% of schools have toilets accessible to students with limited mobility [5]. The lowest toilet coverage in schools were reported in Tanzania (11%), Niger (14%) and Congo (15%) [6]. South Africa on the other hand had the highest coverage of (100%). Rwanda and Ghana had (95%) and (62%) coverage respectively [6]. Between 2010 and 2016, 14 countries recorded at least a five-percentage point decrease in the proportion of schools with no sanitation service [6]. The Democratic Republic of the Congo and Lao PDR had reductions of 19 and 18 percentage points respectively and Peru, Ghana, Gambia and Burundi all succeeded in reducing the proportion of schools with no service to less than 10%.

It has however been observed that the availability of sanitation facilities does not necessary result in its use. [7] also observed that, in some places where toilet facilities were available, open defecation was practiced. These findings lend credence to the fact that, a user's decision to openly defecate maybe influenced by technological, socio-cultural and behavioral factors. In some areas of India and East Java in Indonesia for instance, open defaecation is a social norm issue and is strongly nurtured by allocation of sites for it [8]. Other studies also revealed that in some ethnic cultures, traditional beliefs such as a father-in-law and a daughter-in-law cannot use the same toilet, or where menstruating women are banned from toilet use with a belief that they are untouchable during those menstruating days [9] are situations that compels the practice of open defaecation. A study conducted on open defaecation in rural communities to determine the cultural factors that reinforced its practice in four West African countries—Burkina Faso, Ghana, Mali, and Nigeria showed that the practice of open defaecation was surrounded by cultural taboos and beliefs and linked to ethno-linguistic groups [10]. [11] also found from his study that people imitate others who defaecate indiscriminately thinking that it is good practice. [12] in their systematic review of literature on how different sanitation interventions impact latrine coverage and use, found from their study that, most sanitation interventions only had a modest impact on increasing latrine coverage and use.

The main objective of this study is to understand the cultural issues surrounding the practice of open defaecation and help shift culturally induced open defaecators to sustainable toilet users' culture and open defaecation. This paper provides an analysis of specific cultural variables and their influence open defaecation behaviour within the basic school system. It is anticipated that, the study results will contribute to policy direction towards reducing and preventing open defaecation especially in the school setting both in Ghana and globally.

## Methods

### Conceptual framework

The conceptual framework guiding the conduct of this study is adapted from [13]. This framework postulates that open defaecation is influenced by intrinsic individual behaviour which depends on attitude, subjective norms and perceived behavioural control [14], and other external factors such as the availability or otherwise of physical facility.

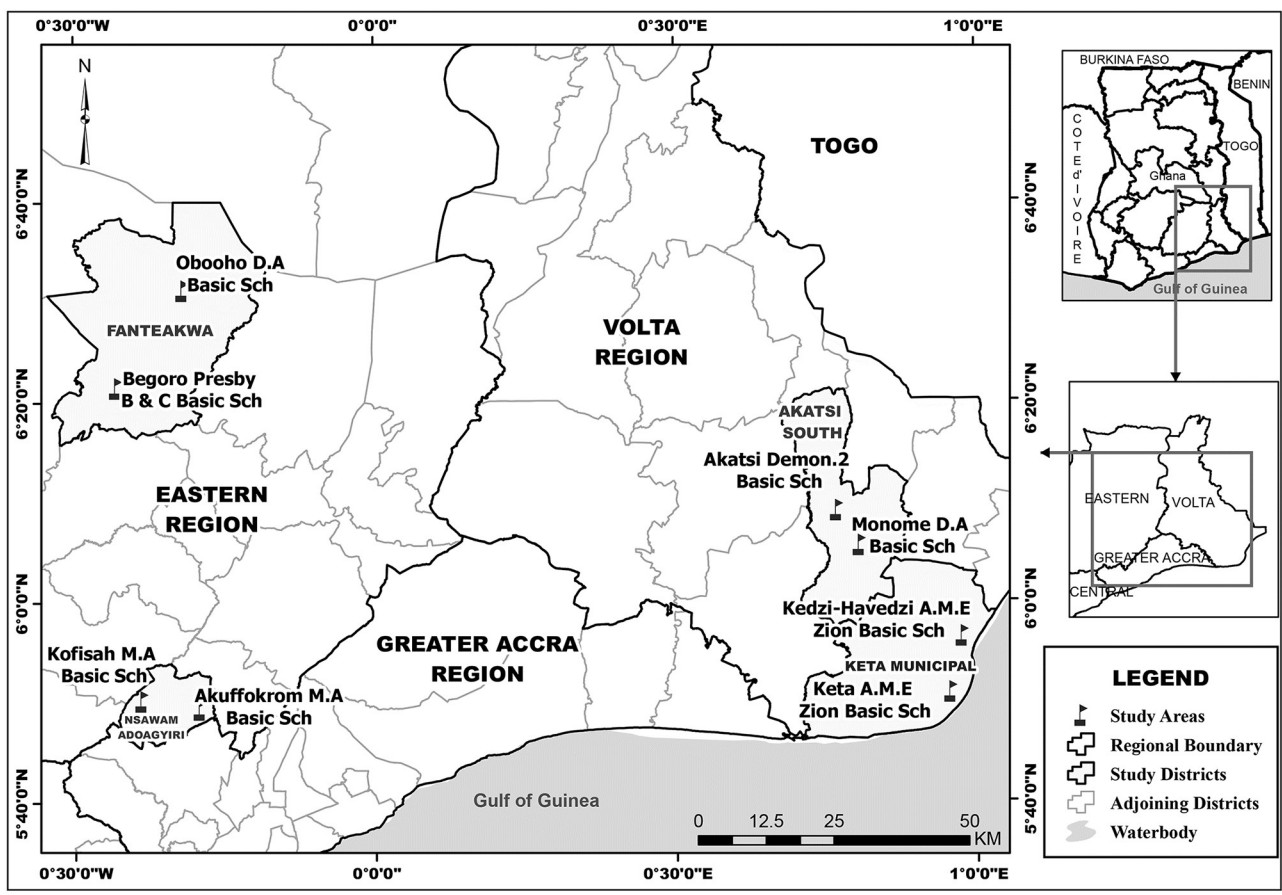

**Fig 1. Map showing study districts and schools.**

### Study area

The study was a school-based cross-sectional study that was conducted between February to April 2016 in the Eastern and Volta Regions of Ghana. Two municipal/district assemblies were selected from each Region (Fig 1). These municipality/ districts were purposively chosen based on previous reports of high open defaecation prevalence rates [15]. In the Eastern Region, the Fanteakwa district, located on longitudes 0˚32.5' West and 0˚10' East and latitudes 6˚ 15' North and 6˚ 40' North and Nsawam- Adoagyiri municipal assembly, located on latitude 5'.45 N and 5'.58 N and longitude 0.07'W and 0.27' W were selected. In the case of the Volta Region, Akatsi South district which is located between latitude 6˚ S—7˚ North 0˚ W—1 ˚ W and Keta municipal assembly, located between- Longitudes 0.30E and 1.05E and Latitudes 5.45N and 6.005N were selected.

### Study communities and schools

Within each municipality/district, two communities were randomly selected from which one public school was selected from each of the communities. The chosen communities in Fanteakwa were Begoro and Oboaho and from Nsawam—Adoagyiri, Kofisah and Akuffokrom. Residents of these communities are predominantly Akans and mainly engage in farming. In the Keta municipality, Kedzi-Havedzi and Keta were the communities from which study schools were selected and in Akatsi South district, the selected schools were in Akatsi, and

Monome. Majority of the residents of the Keta municipality and Akatsi South district are fishers, farmers and traders. In all, a total of eight public basic schools were selected for the study. Where the community has more than one public school, simple random sampling was used to select the school. Inclusion criteria for selected schools were schools with basic sanitation facilities. Schools without toilets were excluded from the study. All studied schools had Kumasi Improved Ventilated Pit (KVIP) latrines. The average population of pupils in the study schools in Fanteakwa and Nsawam- Adoagyiri was 248 and Keta municipality and Akatsi South district was 308. The selected communities Oboaho, Kofisah, Kedzi-Havedzi, and Monome are rural communities (community with population <5000 people), Begoro, Akuffokrom, Keta, and Akatsi are urban settings (with population >5000 people).

## Study participants and sample size determination

The study participants were made up of pupils between the ages of 9 and 18 years. These were children in the upper primary (classes 4, 5 and 6) and Junior High School (JHS) making up classes from forms 1, 2, 3. Children in the lower primary school were excluded from the study because of inability to articulate their thoughts and also understand the items on the questionnaire. The enrollment of the study participants was upon provision of verbal consent from their parents, teachers and the pupils and participation in the study was voluntary.

The sample size for the study was determined using the statistical formula stated below [16]:

$$n = \frac{Z^2 PQ}{d^2}$$

Where,

n = required sample size,

Z = Reliability coefficient for 95% confidence level usually set at 1.96.

P = proportion of the population having the desired characteristics. As a rule of thumb [worst case scenario], 50% was used because there was no reasonable estimate of pupils who defaecate in the open in literature reviewed.

Using the formula, a total sample size of 384 was estimated. This was adjusted for a likely non-response rate of 4% thus increasing the sample size to from 384 to 400 pupils. The pupils who expressed willingness to participate in the study were grouped into males and females. Based on a total sample size of 400, fifty pupils were selected from each of the eight (8) study schools. Each group was made to pick a number from a set of numbers ranging from 1 to 50. Only pupils who picked the numbers 1 to 25 from each of the groups were enrolled in the study. Male and female pupils who picked numbers above 25 to 50 were excluded from the study.

## Data collection

A mixed method approach comprising the use of a structured questionnaire and focus group discussion (FGD) were employed for the study. The primary measure for open defaecation was defined as practice of open defaecation in past one week prior to the survey. The one-week recall was based on the premise that the prevailing environmental sanitation situation in the study areas will have been the same as the previous one week. The selected cultural variables that were assessed for their influence on open defaecation practice were based on literature, expert's opinion and also drawn from interactions with the study community members during reconnaissance visits. The cultural themes emerging from the various engagements were categorized as superstition, taboos, traditional norms and traditional value (Table 1).

**Table 1. Categorization of cultural variables.**

| Culture variable | Sub-culture variable | Categorization |
|---|---|---|
| **Superstition** | Defaecating in toilet causes one to be possessed by evil spirits. | **[S1]** |
| | Shared toilets are associated with evil spirits and therefore should avoided. | **[S2]** |
| | Using same toilet with people in the other houses causes one to lose his/her social status. | **[S3]** |
| | Defaecating in public toilet causes one to lose his/her magical powers. | **[S4]** |
| **Traditional norms** | Faeces are not things to be kept in homes/ in toilet in the house. | **[T1]** |
| | Girl who are in their menstrual periods are not supposed to defecate in toilet. | **[T2]** |
| | Children can defecate in the open since their feces are considered harmless. | **[T3]** |
| **Traditional value** | Defaecating in the open (bush, beaches) signifies continuation of ancestor's way of life. | **[TV1]** |
| | Contact with human faeces is unacceptable. | **[TV2]** |
| *Taboo | Defaecating in an enclosed place (inside toilet) is a taboo). | **[TB]** |

*Taboo did not emerge as a major theme from the interaction with community members during reconnaissance visit.

A pre tested structured questionnaire was developed and this was tested using the Cronbach's Alpha reliability test for the internal consistency of the questionnaire. The questionnaire captured data on demographics, knowledge awareness, knowledge and perception of open defaecation and cultural issues. Out of those administered the questionnaire a sample of was drawn for the FGDs. The FGDs made up of same sex groupings to create a comfortable environment for open discussions. Four (4) FGDs sessions (FGDs) comprising two for female pupils and two for male pupils were conducted in each of the eight studied schools. Each FGD consisted of 6 to 9 pupils.

Two weeks prior to the data collection, the two research assistants were recruited and trained to administer the questionnaire as well as trained on how to translate concepts into the main local dialects spoken in the study communities. Although the survey was administered mainly in English, periodically, the local dialects (such Twi and Ewe) were used to clarify information being communicated when required. Focus group discussion lasted for 30 to 45 minutes. All the FGDs took place at the school compound.

## Data analysis

The analysis of the study was based on the following standard definitions:

**Open defaecation.** Open defaecation involves depositing human excreta outside designated place such as fields, forests, bushes, bodies of water [17]. For the purpose of this study, open defaecation was defined as depositing human excreta outside toilets for at least once within seven days prior to data collection.

**Behaviour.** Behaviour is every action by a person that can be seen or heard [18].

**Culture.** This refers to knowledge, attitude, beliefs, morals, customs and any other capabilities and habit acquired by a person as member of society [19]. It defines what people think and do and everything they have as members of society [20].

**Quantitative analysis.** Statistical Package for Social Sciences (SPSS) Version 20 (IBM) Software (SPSS Inc. Chicago, IIinois, USA) and Analysis of Moment Structures (AMOS) software version 24 were used to analyze the quantitative data. Missing data was not considered in the analysis. The level for analysis was done at the district level. Frequency was used as descriptive statistic to analyze the responses and chi-square test were used to test for association

between OD practices and culture. Culture was assessed using sub- variables which were pooled together for the analysis. The sub-cultural variables were assessed on a Likert scale as strongly agree, agree, don't know, disagree and strongly disagree. To facilitate analysis, the responses were regrouped numerically as strongly disagree and disagree = 0; strongly agree and agree = 1 and don't know not included in analysis. The pooled sub-cultural variables were analyzed for the cultural effect. All analysis was done at 95% confidence limit.

Based on the chi square test, variables showing association were further analyzed using the structural equation modeling (SEM) path. The SEM was employed because of its robustness in controlling for measurement errors associated with collinearity and outliers as described in [21]. The Theory of Planned Behaviour [22] formed the theoretical background for which the proposed interaction between open defaecation intention and practice, and the cultural variables were assessed. Four SEM path models $M_1$, $M_2$, $M_3$ developed. The $M_1$-$M_2$ path models were applied to estimate the relative mean moderating effect sizes of the predicted variables (superstition, traditional norm, traditional value and taboo) and the pooled cultural variables respectively, and $M_3$ showing the moderating effects of the demographic variables. The measurement scale of the effect size was set within the range of 0–1. Negative effect sizes are considered not to influence open defaecation intention and practice.

The general structural equation model as outlined by [23] and presented below was used to calculate the moderating effect size of the cultural factors (superstitions, taboos, traditional norms and traditional values) on pupils' open defaecation behaviour.

$$\boldsymbol{\eta} = \mathbf{B}\boldsymbol{\eta} + \mathbf{I}\zeta + \zeta \tag{1}$$

Where $\boldsymbol{\eta}$ is a vector of sub-cultural variables, $\boldsymbol{\xi}$ is a vector of independent cultural variables, $\mathbf{B}$ is a matrix of regression coefficients relating the sub-variables, I is a matrix of regression coefficients relating sub-variable to the cultural variable, and $\zeta$ is a vector of disturbance terms.

The sub-variables are linked to open defaecation via measurement equations for the sub-cultural variables and cultural variables. These equations are defined as:

$$\mathbf{y} = \boldsymbol{\Lambda}_\mathbf{y}\boldsymbol{\eta} + \mathbf{e} \tag{2}$$

and

$$\mathbf{x} = \boldsymbol{\Lambda}_\mathbf{x}\mathbf{y}\boldsymbol{\xi} + \partial \tag{3}$$

where $\boldsymbol{\Lambda}$y and $\boldsymbol{\Lambda}$x are matrices of factor loadings, respectively, and $\mathbf{e}$ and $\partial$ are vectors of uniqueness, respectively. In addition, the general model specifies variances and covariances for $\boldsymbol{\xi}$, $\zeta$, $\mathbf{e}$, and $\partial$, denoted $\Theta$, $\Psi$, T e, and T, respectively.

**Qualitative analysis.** Data generated from FGDs was transcribed verbatim into Microsoft Word and analyzed using the thematic analytical procedure outlined in [24]. A thematic framework was developed based on identifying texts with similar meaning into themes. The themes were then categorized into the four broad areas of cultural interest i.e. superstitions, traditional norms, traditional values and taboos. Descriptive accounts of the participants' views were exactly listed and from which meanings were inferred from the researchers' understanding.

## Ethics

Ethical permission to undertake the study was granted by Ethical Committee for College of Basic and Applied Sciences (CBAS), University of Ghana (ECBAS 035/15-16). Permission was also sought from the Ghana Education Service (GES), Municipal and Districts Education Directors in charge of basic schools' head of schools and pupils. The research team also sought

**Table 2. Demographic profile of study participants.**

| Characteristic | Eastern Region | | Volta Region | | Total |
| --- | --- | --- | --- | --- | --- |
| | Fanteakwa | Nsawam-Adoagyiri | Akatsi | Keta | |
| | Respondents (%) | Respondents (%) | Respondents (%) | Respondents (%) | Respondents (%) |
| **Sex**** | | | | | |
| Male | 54(54.0) | 60 (60.0) | 36 (36.0) | 50 (50.0) | 200 (50) |
| Female | 46 (46.0) | 40 (40.0) | 64 (64.0) | 50 (50.0) | 200 (50) |
| Chi square (p-value) | | | | | |
| 19.7 (0.003) | | | | | |
| **Age Group** | | | | | |
| 9–11 | 12 (12.0) | 7 (7.0) | 20 (20.0) | 20(20.0) | 59 (14.8) |
| 12–15 | 39 (39.0) | 48 (48.0) | 55 (55.5)) | 28 (28.0) | 200 (50.0) |
| 16–18 | 49 (49.0) | 45(45.0) | 25 (25.0) | 22 (22.0) | 41 (35.2) |
| *Mean age.15.7; SD = ± 2.056* | | | | | |
| *Chi-square (p-value): 44.1 (0.830)* | | | | | |
| **Educational level** | | | | | |
| P4-P6 | 45 (45.0) | 51 (51.0) | 36 (36.0) | 60 (60.0) | 192 (48.1) |
| JHS 1- JHS 3 | 55 (55.0) | 49 (49.9) | 64 (64.0) | 39 (39.0) | 207 (51.9) |
| Chi-square(*p-value*) 43.8 (0.173) | | | | | |
| **Religion** | | | | | |
| Christian | 92(92.0) | 88(88.0) | 88(88.0) | 88(88.0) | |
| Islam | 5(5.0) | 10(10.0) | 9(9.0) | 12(12.0) | - - |
| Traditional | 3 (3.0) | 2(2.0) | 3 (3.0) | 0(0.0) | |
| *Chi square (p-value)* | | | | | |
| *35.8 (0.602)* | | | | | |
| **Ethnicity** | | | | | |
| Ga | 6(6.0) | 22(22.0) | 6(6.0) | 5(5.0) | 39 (9.8) |
| Ewe | 52(52.0) | 28(28.0) | 55(55.0) | 81(81.0) | 216 (54.0) |
| Akan | 28(28.0) | 47(47.0) | 38(38.0) | 9(9.0) | 122 (30.5) |
| Ga Dangme | 12(12.0) | 3(3.0) | 0(0.0) | 1(1.0) | 16 (4.0) |
| Others | 2(2.0) | 0(0.0) | 1(1.0) | 4(4.0) | 17 (1.8) |
| *Chi square (p-value)* | | | | | |
| *20.1 (0.025)* | | | | | |

Numbers in parenthesis showed percentage frequency distributions.

** Fifty percent (50%) of the total number of participants were females and 50% were males.

permission from the chiefs of the study communities. Through the head of schools, parents, were informed of the research and permission sought from them to allow their children to participate in the study. Recruitment and participation in the study was voluntary and subjects could withdraw at any stage of the data collection process without any negative consequences.

## Results

### Characteristics of study participants

Four hundred pupils were administered the questionnaires. These were made up of 200 males and 200 females. The demographic profile of the study participants is presented in Table 2. In Fanteakwa and Nsawam, there were more males (54.0% and 60.0% respectively) than females' participants whereas Akatsi had more females (64.0%) than males (36.0%) and Keta had equal

**Table 3. Occurrence of open defaecation among pupils.**

| Characteristic | Pupils who open defaecate / Total respondents | Chi -Square (P value) |
|---|---|---|
| **District** | | |
| Fanteakwa | 47/257(18.3%) | 5.7 (0.12) |
| Nsawam-Adoagyiri | 71/ 257 (27.6%) | |
| Akatsi South | 71 / 257(27.6%) | |
| Keta | 68/257 (26.5%) | |
| **Gender** | | |
| Male | 119/257 (46.3) | 5.8 (0.32) |
| Female | 138/257 (53.7) | |
| **Age group** | | |
| 9–11 | 43/257 (16.7%) | |
| 12–15 | 130/257 (50.6) | 44.1(0.83) |
| 16–18 | 84/257 (32.7) | |
| **Setting** | | |
| Urban | 117/257 (45.5%) | 14.38 (0.13) |
| Rural | 140/25 (54.5%) | |
| **Class** | | |
| Upper Primary | 129/256 (50.4) | |
| JHS | 12/256 (49.6) | 43.8 (0.17) |

representation of females and males. The mean age of the study participants was 15.7 years and majority of the pupils were in JHS. Most of participants were Christians.

## Open defaecation practice

Occurrence of OD was found to be high in all the study schools (Table 3). Out of a total of 347 respondents, 257 (74.1%) of them practice open defaecation. Interestingly more females (53.7%) responded to practicing OD compared to males (46.3%). As expected, OD occurrence was higher in rural schools (54.5%) than urban schools (45.5%). However, there were no significant variation observed between OD occurrence and all the demographic variables.

## Responses to cultural variables tested

The percentage of respondents to cultural variables used to test for the influence of culture on OD is shown in Table 4. Generally, majority of the pupils in all the schools strongly disagreed or disagreed to the fact that culture influenced OD. However, among those who acknowledged that culture influenced OD practice, it was found that superstition, traditional norm and taboo were significantly associated with OD practice (Table 4).

Table 5 presents the association between OD and demography and culture. It was found from the study that, among those who acknowledged the influence of culture, age group was associated with superstition (all three sub-variables), traditional norms (T2 and T3) and traditional value (TV2) (Table 5). More of the youngest of pupils (9–11 years) and those between classes 4–6 acknowledged the influence of superstition (Defaecating in toilet causes one to be possessed by evil spirits (S1) and Shared toilet is associated with evil spirits and therefore should avoided (S2) on OD compared to older pupils > 11years and those in the JHS respectively.

Regarding the urban rural settings, most the variables did not show significant variation except for the acceptance of children defaecation in the open (T3), which was found to be high among pupils in urban schools compared to rural schools, whereas more rural schools

**Table 4. Acknowledgement of cultural factors influencing open defaection.**

| Variables | Code | Sub-variables | No. of pupils acknowledging theme/ Total respondents | Percent response % | Chi square | P value |
|---|---|---|---|---|---|---|
| **Superstition** | [S1] | Defaecating in toilet causes one to be possessed by evil spirits | 36/307 | 11.7 | | |
| | [S2] | Shared toilet is associated with evil spirits and therefore should avoided. | 41/316 | 13.0 | | |
| | [S3] | Defaecating in public toilet causes one to lose his/her magical power | 50/302 | 16.6 | | |
| *Pooled score* | | | | *13.5* | 60.4 | 0.00 |
| **Traditional norms** | [T1] | Faeces are not things to be kept in homes/ in toilet in the house. | 120/324 | 37 | | |
| | [T2] | Girl who are in their menstrual periods are not supposed to defecate in toilet. | 65/307 | 21.2 | | |
| | [T3] | Children can defecate in the open since their feces are considered harmless. | 55/323 | 17 | | |
| *Pooled score* | | | | 25.1 | 40.4 | 0.019 |
| **Traditional value** | [TV1] | Defaecating in the open (bush, beaches) signifies continuation of ancestor's way of life. | 149/329 | 45.3 | | |
| | [TV2] | Contact with human faeces is unacceptable. | 84/344 | 24.4 | | |
| | [TV3] | Using same toilet with people in the other houses causes one to lose his/her social status | 78/317 | 24.6 | | |
| *Pooled score* | | | | 31.4 | 23.3 | 0.5 |
| **Taboo** | [TB] | Defaecating in an enclosed place (inside toilet) is a taboo). | 41/326 | 12.6 | 39.3 | 0.028 |

acknowledged that sharing household toilets with others could result in one losing their social status (TV3). There was no association found between the demographic variables and taboo as well as between gender and all cultural variables.

The strength of the relationship between the cultural variables that were found to be associated with open defaecation behaviour using the SEM model is presented (Table 6). Out of the variables assessed, superstition produced the largest effect size (r = 0.728; p < 0.05) followed by taboo (r = 0.46) and traditional norm (r = 0.43).

**Qualitative study.** Four FGDs in each of the 8 schools making a total of 32 focus group discussions was conducted. This was made up 16 male groups and 16 female groups with an average group size between 6–9 participants. In all 192 pupils, made up of 87 females and 105 males participated in the FGDs.

Findings from the focus group discussion corroborate that of the quantitative study (Table 7). It was observed that most of the reasons provided by the pupils for the practice of OD were related to poor environmental sanitation condition of the school toilets. Most pupils avoided toilet use because of uncleanliness of the place, defaecation around the drop hole and the fact that the place smells. Other issues also raised were privacy, having to queue to use the toilet and safety. Only a few cultural factors were raised in the discussions the (Table 7). Examples of cultural views expressed are presented below.

I *don't like to use the toilet because I am afraid of being possessed by evil spirit"*.

(Discussant: FGD.)

"*I defaecate in the open because even our chief also does it"*.

(Discussant: FGD.)

**Table 5. Demographic factors associated with selected cultural variables.**

| Demographic characteristic | Defaecating in toilet causes one to be possessed by evil spirits (S1) | Shared toilet is associated with evil spirits and therefore should avoided (S2) | Defaecating in public toilet causes one to lose his/her magical power (S3) | Girl who are in their menstrual periods are not supposed to defecate in toilet. Girl who are in their menstrual periods are not supposed to defecate in toilet (T2) | Children can defecate in the open since their feces are considered harmless(T3) | Contact with human faeces is unacceptable (TV2) | Using same toilet with people in the other houses causes one to lose his/her social status (TV3) | Defaecating in an enclosed place or inside toilet is a taboo |
|---|---|---|---|---|---|---|---|---|
| **Gender** | | | | | | | | |
| Male | | | | | | | | |
| Female | | | | | | | | |
| **Age group** | | | | | | | | |
| 9–11 | 24.5 | 26.4 | | 30.6 | | 43.9 | | |
| 12–15 | 9.7 | 18.6 | | 27.8 | | 23.5 | | |
| 16–18 | 7.8 | 7.7 | | 8.9 | | 27.1 | | |
| | Chi 10.8 | 11.2 | | 18.3 | | P = 9.2 | | |
| | p = 0.005 | p = 0.004 | | p = 0.00 | | 0.01 | | |
| **District** | | | | | | | | |
| Fanteakwa | | | 6.9 | 14.3 | 5.3 | 43.4 | 38.5 | |
| Nsawam-Adoagyiri | | | 12.5 | 9.8 | 20.0 | 38.3 | 48.9 | |
| Akatsi South | | | 33.3 | 38.4 | 28.1 | 45.2 | 28.1 | |
| Keta | | | 13.3 | 23.4 | 11.2 | 59.4 | 31.6 | |
| | | | Chi = 24.5; p = 0.00 | Chi = 24.5. p = 0.00 | Chi = 20.7 P = 0.00 | Chi = 9.7, p = 0.025 | Chi 9.8, p = 0.02 | |
| **Setting** | | | | | | | | |
| Urban | | | | | 23.2% | | 38.2 | |
| Rural | | | | | 8.9 | | 54.6 | |
| | | | | | 14.2 P = 0.00 | | 10.4 P = 0.001 | |
| **Class** | | | | | | | | |
| 4–6 | 16.6 | 20.8 | | | | | 36.8 | |
| JHS 1–3 | 7.3 | 11.6 | | | | | 55.3 | |
| | 9.7P = 0.002 | 5.3 P = 0.02 | | | | | 13.0 P = 0.0 | |

* Grey shaded boxes depict insignificant association between demographic variable and culture variables.

**Table 6. Moderating effect size of culture and demographic variables on open defaecation behaviour.**

| SEM Path Model | Cultural/Demographic Factors | Standardized Effect Size | S.E. | p-value |
|---|---|---|---|---|
| $M_1$ | Superstitions | 0.728 | 0.180 | 0.00 |
| | Traditional norms | 0.425 | 0.105 | 0.00 |
| | Taboos | 0.462 | 0.115 | 0.00 |
| $M_2$ | Pooled Cultural Factors | 0.23 | 0.252 | 0.021 |
| $M_3$ | Gender | -0.028 | 0.198 | 0.571 |
| | Age | -0.021 | 0.057 | 0.602 |
| | Education | -0.038 | 0.046 | 0.447 |
| | Setting | 0.101 | 0.188 | 0.042 |

**Table 7. Selected quotes from focus group discussions.**

| Quote | Issue |
|---|---|
| • *"...The toilet is not clean and when you go there, you will see faeces and anal cleansing materials scattered on toilet floor"*—(Female pupil.) | ES |
| • *"...It is not neat because people from the town smoke there and also ease on the squat holes and footrests"*—(Female pupil, FGDs.) | |
| • *"Sometimes, they urinate on the toilet floor and defaecate around it making it dirty"*—(Male pupil, FGD.) | |
| • *"...Sometimes you get pressed with the faeces but getting there all cubicles are occupied."*—(A girl, FGD.) | A |
| • *"If you go there and the toilet is full you have to wait; if you can't wait, you have to be shouting "I want to defaecate"*—(Male pupil, FGD.) | A |
| • *"We queue to use toilet."*—(Female pupil, FGD.) | |
| • *"I don't feel like defaecating again when they come and knock the door"*—(Male pupil, FGD.) | P |
| • *"The inner lockers are spoilt as such someone can open the door and see your private parts."*—(Female pupil, FGD.) | |
| • *"The doors are not good, so I don't have enough privacy in the toilet; you can be seen by anyone who comes there."*—(Female pupil, FGD.) | P |
| • *"...The toilet smells and you have to remove your uniforms before defaecating in the toilets*—(Male pupil, FGD.) | ES |
| • *"...When we are asked to go and clean the toilets, we can't go there because the toilet smells"*—(Male pupil, FGD.) | |
| • *I don't go there because the scent of the faeces stays in my dress"*—(A female pupil, FGD.) | ES |
| • *"Pupils under five years using same squat holes as adults; they can fall into the pit"*—(Male pupil, FGD.) | ES |
| • *"The preschool children also used our toilets and defaecate on the squat holes and make the place dirty"*—(Male pupil, FGD.) | A |
| • *"...we need nose masks and gloves so we can clean the toilet"*—(Female pupil, *FGD.)* | ES |
| • *". We need chamber pot for the preschool children, so they don't make the toilet dirty by defaecating on the floor."*—(Female pupil, FGD.) | S |
| • *"We need detergents to clean the toilet"*—(Female pupil, FGD). | ES |
| • *"...all we want is that the community should be told not to defaecate on the floor"*—(Female pupil, FGD.) | |
| • *". They make the toilet filthy and full quickly; they also soiled squat holes together with the feet rests; others leave their sanitary pads in the toilet"*—(Male pupil, FGD.) | ES |
| • *"When school closes in the afternoon, we locked the toilet doors; but the town people come and break the padlocks and defaecate in the toilet"*—(An 18-year Female pupil, FGD.) | ES |
| • *"When I go to the toilet and see rodents, I don't feel comfortable and I cannot even ease myself properly"*—(Male pupil, FGD). | S |
| • *Avoided toilet use because of fear of being possessed by evil spirit".* (Discussant: FGD.) *** | Su |
| • *"Defaecate in the open because the chief of my community also does it".* (Discussant: FGD.) *** | TN |
| • *"Even though I have toilet in the house, I feel more comfortable defaecating in the bush"* (Discussant: FGD.) *** | PN |

Key issues from FGD categorized as ES- environmental sanitation, A -access, P-privacy, S-safety, Su-superstition, TN -traditional norm, PN-personal norm.

*** -cultural factors mentioned.

## Discussion

In the pre-colonial traditional African societies, the environment was viewed with a metaphysical outlook. This metaphysical view underpinned why traditional Africans were more cautious in their attitude to plants, animal and inanimate things and the various invisible forces of the world [25]. Within the African metaphysical worldview, there is a slim dichotomy between "plants, animals, and inanimate things; between the sacred and the profane; "communal and the individual" and "matter and spirit" [25]. It is in line with this metaphysical framework that one can consistently and coherently situate the people's belief in transmigration of the soul/

spirits into animals, plants or into forces such as the wind as an example. It is however noted that many contemporary African and especially urban societies have evolved significantly leaning towards economic development to define their way of life.

To this end, many studies conducted on open defaecation have been geared towards understanding issues on knowledge, availability, access to sanitation facilities and economic factors influencing sanitation behaviour. Sanitation behaviour are also studied broadly and hence the need to understand the influence of culture on open defaecation is important in contributing to finding sustainable solutions to sanitation.

## The influence of culture on open defaecation practice

This study found majority of the pupils did not think culture was an important factor that influenced the OD practice, it cannot be said to be the outright case as minority of the pupils considered culture as an influencer of OD practice anyway. The fact that culture is considered by some to the pupils to influence OD is consistent with many other studies that have identified cultural factors as playing a role in OD practice [26, 27]. For instance, studies in rural settings in Ghana, Burkina Faso, Mali, and Nigeria, have found that taboos, beliefs and values are major cultural factors influencing open defaecation behaviour [27].

We however find literature to be limited on granular studies on culture and OD behaviour and that our study is one among the few that is granular in this context. Based on the moderating effect size determined in the study, superstition was found to be 70% more likely to influence OD behaviour compared to 46% for taboo and 42% for traditional norms.

Many cases of cultural intolerance for the handling of faeces have been associated with superstitious beliefs and reported in countries such as China, India, and Ghana [28–30]. In Sierra leone in West Africa for instance, it is believed that one should not sit over someone else's faeces because it will lead to bad luck [31] and in Uganda, (East Africa) it is believed that pregnant women should not use the toilet because of fear of the death of the foetus [32]. In the case of this study the superstitious beliefs about open defaecation are related to entry evil spirits into the user and loss of social status and magical powers.

The case of taboo being associated with OD behaviour in this study was unexpected and cannot be explained as it did not come out as a major cultural factor during the reconnaissance visits and the FGDs. Nevertheless, other studies have found men to refuse to stop open defaecation because of toilet-associated taboos with in-laws and female grown-up children [33].

The fact that traditional norms have been suggested to influence OD has been reported in previous studies. A similar study conducted in Northern Ghana found 57% of adults acknowledge that OD was an 'age long practice" compared to other reasons such as financial constraints (18.6%) and bad condition of public toilets (8.4%) [33]. Unlike the case of Osumanu et al.'s study which focused on adults, this study was conducted among the school aged children. It is however suggested that children practice what they learn from their communities and household settings [34, 35], and hence what is observed in the school environment may be inferred from what happens in their communities. It is also suggested that perceived social sanitation norms may enhance emotional satisfaction where open defaecation practice is rampant and negate effort to improve sanitation behaviour [35]. The absence of policy directives targeted at open defaecation can also be associated with OD practice in the schools.

## Non-cultural influences on open defaecation

Aside the cultural factors which were the main focus of this study, this study also revealed that the main reasons ascribed to the non-use of school toilets had to do with access, privacy, safety and cleanliness issues. This confirms similar observations made by [36] who found poor

sanitation linked to the absence of toilet facilities or few available toilets for more people. Also [33], reported school children reporting non-use of toilets because they were smelly and dirty.

## Study limitations

We consider some limitation for which our findings may be used with caution. Firstly, the number of schools studied is not large to be representative of schools in the study districts and that all selected schools were public schools. Secondly, did not validate findings from parents or even community members. Thirdly, the pupils were not interrogated for their toilet use behaviour at home to consolidate the practice and to make strong linkages with other factors that may also influence OD practice such as access to household toilets and water supply. It is therefore suggested that further studies be conducted on a wider scale to present a more robust situation in the districts.

## Conclusion

This study established high occurrence of OD in all the selected study schools which does not augur well for teaching, learning and the health of stakeholders in the schools. Culture was found to influence OD behaviour. Superstition, traditional norms and taboo were seen to predict OD practice. To reduce OD practice in basic schools, the study recommends that, the Ministries of Sanitation and Water Resources and Education should champion the development of educational campaigns for good sanitary practices at the national and district levels. The messaging for these campaigns should resonate with the target audience by making them culturally sensitive that will lead to better acceptance and adoption of appropriate sanitation behaviour. The campaigns can be implemented using various platforms such as mass media, social media, school clubs and community groups to reach key stakeholders such as teachers, parents, school children traditional leaders and the general population.

Also, Ministry of Education should update its basic school curriculum to include subjects and themes on open defaecation as a way to teach pupils about the open defaecation and its negative consequences. The local cultural beliefs systems relating to sanitation should be integrated into the curriculum.

## Supporting information

**S1 Data.**
(SAV)

## Acknowledgments

The authors are grateful to the staff and pupils of the study schools for their contribution to the success of this study. The authors are equally grateful to Mr. Dagadu and Ernest Nkansah for supporting with statistical analysis, the study communities, heads of schools, teachers and the pupils for their participation in the study.

## Author Contributions

**Conceptualization:** Saviour Victor Adjibolosoo, Benjamin Dankyira Ofori, Philip Baba Adongo, Stephen Afranie, Dzidzo Regina Yirenya-Tawiah.

**Data curation:** Saviour Victor Adjibolosoo, Philip Baba Adongo, Dzidzo Regina Yirenya-Tawiah.

**Formal analysis:** Saviour Victor Adjibolosoo, Philip Baba Adongo, Dzidzo Regina Yirenya-Tawiah.

**Funding acquisition:** Saviour Victor Adjibolosoo.

**Investigation:** Saviour Victor Adjibolosoo.

**Methodology:** Saviour Victor Adjibolosoo, Benjamin Dankyira Ofori, Philip Baba Adongo, Stephen Afranie, Dzidzo Regina Yirenya-Tawiah.

**Project administration:** Saviour Victor Adjibolosoo.

**Resources:** Saviour Victor Adjibolosoo.

**Supervision:** Benjamin Dankyira Ofori, Philip Baba Adongo, Stephen Afranie, Dzidzo Regina Yirenya-Tawiah.

**Validation:** Dzidzo Regina Yirenya-Tawiah.

**Visualization:** Dzidzo Regina Yirenya-Tawiah.

**Writing – original draft:** Saviour Victor Adjibolosoo, Dzidzo Regina Yirenya-Tawiah.

**Writing – review & editing:** Benjamin Dankyira Ofori, Philip Baba Adongo, Stephen Afranie.

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
