## [Decision Letter · Decision Letter 0]

11 Jun 2020

PONE-D-20-14602

The Influence of Culture on Open Defecation in Selected Basic Schools in Ghana

PLOS ONE

Dear Dr. Yirenya-Tawiah,

Thank you for submitting your manuscript to PLOS ONE. After careful consideration, we feel that it has merit but does not fully meet PLOS ONE’s publication criteria as it currently stands. Therefore, we invite you to submit a revised version of the manuscript that addresses the points raised during the review process.

We look forward to receiving your revised manuscript.

Kind regards,

Khin Thet Wai, MBBS, MPH, MA (Population & Family Planning Resear

Academic Editor

PLOS ONE

Journal Requirements:

2. We note that Figure 2 in your submission contain map images which may be copyrighted. All PLOS content is published under the Creative Commons Attribution License (CC BY 4.0), which means that the manuscript, images, and Supporting Information files will be freely available online, and any third party is permitted to access, download, copy, distribute, and use these materials in any way, even commercially, with proper attribution. For these reasons, we cannot publish previously copyrighted maps or satellite images created using proprietary data, such as Google software (Google Maps, Street View, and Earth). For more information, see our copyright guidelines: http://journals.plos.org/plosone/s/licenses-and-copyright.

Additional Editor Comments (if provided):

General comment:

Extensive English language correction is deemed necessary

throughout the manuscript.

LINE 67 and 77: to correct the duplicate of mentioning 'aspects of the larger study' and also needed to cite a reference for that larger study. Has it been published elsewhere?

LINE 77: to clarify 'sought to understand other behavioural

factors that influenced OD behaviour' Does it mean 'cultural factors'??

To add the sample size determination and the sampling procedure of the quantitative component

To add details of how qualitative data was collected by FGD by following COREQ guidelines; this component was too sketchy and incomplete.

To clarify how meaningful interpretations on cultural drivers of OD have been done by quantitative and qualitative findings

Reviewers' comments:

Reviewer's Responses to Questions

**Comments to the Author**

1. Is the manuscript technically sound, and do the data support the conclusions?

Reviewer #1: Partly

Reviewer #2: Yes

2. Has the statistical analysis been performed appropriately and rigorously? 

Reviewer #1: Yes

Reviewer #2: Yes

3. Have the authors made all data underlying the findings in their manuscript fully available?

Reviewer #1: Yes

Reviewer #2: Yes

4. Is the manuscript presented in an intelligible fashion and written in standard English?

Reviewer #1: No

Reviewer #2: Yes

5. Review Comments to the Author

Reviewer #1: PONE-D-20-14602

The Influence of Culture on Open Defecation in Selected Basic Schools in Ghana

This manuscript is a mixed-methods study describing factors associated with open defaecation among basic school students in Ghana. While the manuscript is of some interest, I do not think the manuscript in its current form is of sufficient interest and quality to warrant publication in PLOS ONE.

I suggest the authors to look for STROBE check list for cross-sectional studies to ensure reporting is complete and transparent.

https://www.strobe-statement.org/fileadmin/Strobe/uploads/checklists/STROBE_checklist_v4_cross-sectional.pdf

My sense is that the manuscript could be strengthened by several modest changes as outlined below.

General comments

-The authors should use the uniform font style and font size.

-The manuscript is not well-written and needs to be edited by a native English speaker. Necessary to check grammar and spellings.

-Abbreviate first before using short form.

-Note that writing a manuscript is not the same as writing thesis and it should be built not only with concise and clear statements but also all the required information are included.

INTRODUCTION

-Introduction section is not well organized.

-Some sentences are duplicating [E.g. Line 66 and 77]

-Introduction section needs to include What is the problem and its supporting details? What is the added value from the study and why are you conducting it? - Both globally and locally

-The adverse consequences of open defaecation and its supporting data is missing in the introduction.

METHODS

-The whole section of Methods needs to be rewritten. I would suggest the authors to prepare by the STROBE check list. Many items necessary to be described are missing.

-Sampling procedure: The sampling procedure needs to provide more detail. How schools were chosen in the selected districts? How many students were eligible? Any eligibility criteria? How were they selected? Equal proportion for 4 schools? Gender based selection? Residence based selection?

-Data analysis: It is not clear and should be described in detail. The analysis section should reflect fully to the results that you presented in the study. So also, the way you calculated moderating effect size needs to be described.

-Need to explain qualitative part in more detail: FGDs or Interviews? KII? Any criteria for selection?

-What are the operation definition of “Open defaecation intention” and “Open defaecation behavior”?

RESULTS

-The authors tend to cover all information in the text and again in the table. You should only include the most important details in the text when you also have a table.

-Mean age always come with SD.

-Any reasons for stratification of age groups? Age stratification is different by tables – it should be same

-Percentages are not included in all the data.

-I could not find all your cultural variables S1-4, T1-3, TV1-2 and TB in the results section. I only see one question “Defaecating outside toilet signifies continuation of ancestral ways of life” as traditional norms and same applied to T and TV. Does it a composite measure? And the way you calculated should be described in the Methods section.

-Any reasons behind for cross-tabulating cultural variables with background characteristics?

-There is lacking linkage between qualitative and quantitative results.

-Too many tables some of which can be combined. Figure 1 and 3 may not be necessary.

DISCUSSION

-The discussion is weak in light of the findings. The discussion section should be rewritten. The ideas are incoherently mixed together. The discussion needs to focus on the key implications of the data with a separate paragraph for each concept.

-Check the sentence in discussion “Our study showed most of the rural schools to have less quality toilet facilities than the urban schools.” I do not find it in your results.

REFERENCES

-Number in the references were overlapped and as a result, citations and provided references are not matched.

-Why data from 2013 require when updated 2019 data is available? [Check Ref. 2 and 3 by WHO and UNICEF]

Reviewer #2: First of all, I am thankful to editor for inviting me to write the review for this manuscript.

The authors have done a good job and described all section of manuscript in sufficient detail. My suggestions for further improving the manuscript are as follows:

Introduction section is well written. Authors explained the rationale behind their study. However I suggest making this section concise and crispy. Particularly, few details in first three Para may be omitted or concisely written.

Methodology section:

Line 89-91 need not be mentioned in methods as already been mentioned in introduction. There is no mention of sample size estimation and selection of study sample. Justification for using Cronbach’s Alpha reliability test can be given. Details of qualitative data collection is not provided such as subjects and numbers of FGD, Who were interviewed etc. I couldn’t understand Line 161 (The AMOS was development to satisfy the data process demand from research effort)

Result:

Table 3-6 is redundant and should be deleted or reduced in number as information given in running text is more than adequate.

Discussion

The discussion section is well written compared to other sections. Any strength or limitation could have also been mentioned.

6. PLOS authors have the option to publish the peer review history of their article (what does this mean?). If published, this will include your full peer review and any attached files.

Reviewer #1: Yes: Kyaw Lwin Show

Reviewer #2: Yes: Rakesh Kumar

---

## [Author Response · Author response to Decision Letter 0]

3 Sep 2020

PONE-D-20-14602

Authors responses to reviewers’ comments 

Reviewer 1 

PONE-D-20-14602

The Influence of Culture on Open Defecation in Selected Basic Schools in Ghana

AREAS COMMENTS RESPONSES 

GENERAL COMMENTS - The authors should use the uniform font style and font size. Rectified: uniform font (Times New Roman (12 points) was used throughout the entire manuscript as required by the Journal, and indicated in the Authors Guide.

 - The manuscript is not well-written and needs to be edited by a native English speaker. Necessary to check grammar and spellings. Rectified: the entire manuscript has been critically and meticulously assessed and proof-read by English language experts as demanded. 

 - Abbreviate first before using short form. Rectified. OD has been written in full before OD was used in subsequent statements. 

 - Note that writing a manuscript is not the same as writing thesis and it should be built not only with concise and clear statements but also all the required information are included. Critical attention has been placed on the writing of the manuscript dwelling much on the STROBE CHECK LIST.

INTRODUCTION

 - Introduction section is not well organized. This has been re-organised with special emphasis on the following areas as prescribed in the STROBE CHECK LIST.

• The scientific background and rationale for the investigation have been explained and reported accordingly.

• Also, the last sentence of the introduction highlights the specific objective of the study.

 - Some sentences are duplicating [E.g. Line 66 and 77] Cross-checked and synchronized repetitions removed. 

 - Introduction section needs to include:

- What is the problem and its supporting details? The problem has been stated in the introcu

 - What is the added value from the study and why are you conducting it?

- Both globally and locally. This is a granular study that brings out the various cultural elements that influence OD behaviour in the school setting. Not many of this kind of study has been conducted in schools. The results provided an added on knowledge to be considered in developing reducing/preventing strategies for open defaecation in schools both in Ghana and globally.

 - The adverse consequences of open defaecation and its supporting data is missing in the introduction. The adverse consequences has been inputted into the introduction

METHODS

 - The whole section of Methods needs to be rewritten. I would suggest the authors to prepare by the STROBE check list. Many items necessary to be described are missing. 

This was rectified as indicated in the methodology section of the current version of the paper.

 Sampling procedure: 

- The sampling procedure needs to provide more detail. 

- How schools were chosen in the selected districts? 

- Any eligibility criteria? 

- How were they selected? Equal proportion for 4 schools? 

- Gender based selection? Residence based selection? The details of the sampling procedure have been stated in the methods

Eligibility criterion: Schools with functioning toilet were included from the study.

The study did not consider gender based /residence schools. Public schools at the basic school level are usually mixed and non residence in Ghana

 Any eligibility criteria for school selection?

 Yes, stated abovue

 - How many students were eligible? 

- Any eligibility criteria? 

- How were they selected? 

- Equal proportion for 4 schools?

- Gender based selection? The population frame 

Four hundred (400) pupils aged between 9 and 18 years formed the study participants. Those below and above age 9 and 18 respectively were excluded from the study. These participants were selected from eight public first cycle/ basic schools in four districts of the two regions: Eastern and Volta, using simple randomization method. They were in classes 4, 5, and 6 and Junior High School (JHS) forms 1, 2 & 3. 

Exclusion criteria: Pupils in lower primary classes 1, 2, & 3 were excluded from the study because of their inability to articulate their thoughts during the FGDs and also understand the items on the questionnaire guide. 

Sample size determination: The sample size for the quantitative survey was determined using the statistical formula stated below. (Cochran, 1977):

 n = Z2 PQ 

 d2

Where, 

n = required sample size,

Z = Reliability coefficient for 95% confidence level usually set at 1.96.

P = proportion of the population having the desired characteristics. As a rule of thumb [worst case scenario], 50% was used because there was no reasonable estimate of pupils who defaecate in the open in literature reviewed.

This gave a total sample size of 384. 

This, however, was adjusted for a likely non-response rate of 4% thus increasing the sample size to from 384 to 400 pupils. 

 - Gender based selection? Fifty percent (50%) of the participants were female pupils, and 50% were male pupils. 

 - Residence based selection? This was not considered in the investigation as the study was limited to the public basic day schools only.

 - Selection of participants for the qualitative study [FGDs]? The participants for the qualitative study were randomly drawn those who participated in the quantitative study and those who said the OD. In all, data on 192 participants open defaecation behaviours were gathered from the FGD sessions. 

DATA ANALYSIS - It is not clear and should be described in detail. The analysis section should reflect fully to the results that you presented in the study. The data analysis has been addressed to reflect the results presented

All quantitative analyses were carried 95% confident interval

 - So also, the way you calculated moderating effect size needs to be described. The calculation of the moderating effect has been described in the manuscript

 - Need to explain qualitative part in more detail: FGDs or Interviews? KII? 

- Any criteria for selection? The qualitative aspect of the study has been improved. Only focus group discussion was conducted. The criteria for selection for FGD has been stated above 

 - What are the operation definition of “Open defaecation behavior”? For the purpose of this study, open defaecation was defined as depositing human excreta outside designated place such as fields, forests, bushes, bodies of water (WHO/UNICEF, 2008). However, for the purpose of this study, open defaecation was defined as depositing human excreta outside toilets for at least once within seven days prior to data collection.

 - What is the operation definition of “Open defaecation intention” Authors have deleted aspects on intention and focused only on OD practice.

RESULTS - The authors tend to cover all information in the text and again in the table. You should only include the most important details in the text when you also have a table. This was identified and corrected as stated in the reviewer’s comments.

 - Mean age always come with SD. This has been effected in the results

 - Any reasons for stratification of age groups? Age Group stratification was done between the pupils in the Primary and Junior High categories in order to compare the effect size of cultural factors on OD behaviour for the two groups and for policy directions.

 - Age stratification is different by tables – it should be same. The difference in age group stratification was identified and corrected as: 9-11; 12-14; and 15-18 in the tables.

 - Percentages are not included in all the data. The sample size drawn from each school within the District was 100. Hence, by calculation, the percentage responses is equal to the response frequencies for each school/District—Fanteakwa, Akatsi, Keta and Nsawam-Adoagyiri.

 - I could not find all your cultural variables S1-4, T1-3, TV1-2 and TB in the results section. I only see one question “Defaecating outside toilet signifies continuation of ancestral ways of life” as traditional norms and same applied to T and TV. Does it a composite measure? And the way you calculated should be described in the Methods section. Care has been taken to include all cultural variables

The details of how the data was analysed has been improved in the manuscript to cover how the analysis was done

 - Any reasons behind for cross-tabulating cultural variables with background characteristics? This was done to see if there was any variation between OD and demography across the study areas 

 - There is lacking linkage between qualitative and quantitative results. The linkage between the quantitative and qualitative has been made

 - Too many tables some of which can be combined. Figure 1 and 3 may not be necessary Tables had been drastically reduced by summarizing the information in them.

DISCUSSION

 - The discussion is weak in light of the findings. The discussion section should be rewritten. Discussion has been strengthened as indicated below [Additional information from results added=see below]:

 - The ideas are incoherently mixed together. The discussion needs to focus on the key implications of the data with a separate paragraph for each concept. The coherency of the section has been improved to focus on key findings and the implications

REFERENCES

 - Number in the references were overlapped and as a result, citations and provided references are not matched. This has been corrected using endnote

 - Why data from 2013 require when updated 2019 data is available? [Check Ref. 2 and 3 by WHO and UNICEF] The 2013 reference was updated with 2019 reference.

REFERENCES

 - Number in the references were overlapped and as a result, citations and provided references are not matched. Overlapings of references were identified and corrected.

REVIEWER 2 

PONE-D-20-14602

The Influence of Culture on Open Defecation in Selected Basic Schools in Ghana

AREAS COMMENTS RESPONSES

General comments 

INTRODUCTION

 - 

METHODS - Names of study communities and schools. [Setting in parentheses].

 Communities Schools

Kofisah [rural] Kofisah M.A. Primary & JHS

Akuffokrom [urban] Akuffokrom M.A. Primary & JHS

Begoro [urban] Begoro Presby Primary B & C, & JHS

Oboaho [rural] Oboaho D.A Primary & JHS

Keta [urban] Keta A.M.E. Zion Primary & JHS

Kedzi-Havedzi [rural] Kedzi-Havedzi A.M.E. Primacy & JHS

Akatsi [urban] Akatsi Demons. Primary & JHS

Monome [rural] Monome D.A. Primary & JHS

The table above showed names of the study communities. These names were, however, been removed from the text for ethical reasons.

 - Rural & Urban Communities Rural Communities: 

• Kofisah

• Oboaho

• Kedzi Havedzi

• Monome

Urban Communities

• Akuffokrom

• Begoro

• Keta 

• Akatsi

 - Responses of the 45 teachers were not presented in the tables.

- Responses of the 45 teachers for validation. The information (responses) provided by the 45 teachers for validation of pupils views relate to only the conditions of the school toilets and not the cultural factors. This was therefore excluded from the results. 

 - Why not chiefs, opinion leaders, parents, clans men for responses for the validation. These are limitations of the study and has been duly stated in the manuscript

 - Do pupils have in-house toilets facilities to use? This was not factored into the variables of interest and therefore not assessed/investigated by the study. This constituted one limitation of this study.

 - How many pupils have access to in-house toilets facilities? This was not factored into the variables of interest and therefore not assessed/investigated by the study. 

 - How many pupils did not have access to in-house toilet facilities? This was not factored into the variables of interest and therefore not assessed/investigated by the study.

 - It will be good to have data on pupils with in-house toilets and pupils without in-house toilets. This was not factored into the variables of interest and therefore not assessed/investigated by the study.

 - Did you observe adults practicing OD? 

This has not been investigated; however, from the responses of pupils (respondents), it was evident that adults in the communities practice OD. [See FGD results presented in the manuscript. For example one discussant from Kofisah M.A. Basic School explained that: he defaecate in the open because the chief of his community also does it [see FGD results].

 - What is their reasons for disagreeing? 

DATA ANALYSIS: What is the significance of the rural and urban analysis? [See tables 3, 4, 5, 6.] The study intended to compare OD in the rural and localities, hence the inclusion of the two settings. These were generally not signfican and although significance was seen with SEM model the moderating effect was weak

 In exploring the association between demographics and OD & culture, [how many of the pupils have access to in-house toilets; and how many do not? This was not factored into the variables of interest and therefore was not assessed/investigated by the study. Study limitation

DISCUSSION

 How do you reconcile what your study sought to achieve with the works of Osumanu et al. (2019) reference 24. Osumanu et al.,(2019) traditional norms were highly reported by the respondents and this study also found it to be significant

 -Which of your result emphasis this? You did not tell us the state of the school toilet facilities or indicated in your results if the schools studied had access to toilet facilities? This paper is focused on the study of culture and so other factors were given less attention here. The state of school toilets nevertheless have been duly mentioned as it came up during fGD under the qualitative study

 -Also, the issue of rural and urban schools only show in tables 3-6 without any mention in the methodology. Rectified in the method

 Why the rural and urban school comparison? -Which of your results from table 3-9 highlight this sentence? This has been rectified . most of the tables have been removed . It is showing in tables 1. 4 and 5

 Your result is silent on the cleanness of the schools' toilet facilities? The only mention of school toilet facility was in the section 2.2

-The common sanitation facility used in these schools are the Kumasi Improved Ventilated Pit (KVIP) latrines. The study is focused on culture and therefore the seemingly silence on cleanliness. It has nevertheless presented

---

## [Editor Report · Decision Letter 1]

8 Sep 2020

The Influence of culture on open defaecation in some basic schools in selected districts in Ghana: A preliminary study

PONE-D-20-14602R1

Dear Dr. Yirenya-Tawiah,

We’re pleased to inform you that your manuscript has been judged scientifically suitable for publication and will be formally accepted for publication once it meets all outstanding technical requirements.

Kind regards,

Khin Thet Wai, MBBS, MPH, MA (Population & Family Planning Resear

Academic Editor

PLOS ONE
---

## [Editor Report · Acceptance letter]

18 Sep 2020

PONE-D-20-14602R1 

 The Influence of culture on open defaecation in some basic schools in selected districts in Ghana: A preliminary study 

Dear Dr. Yirenya-Tawiah:

I'm pleased to inform you that your manuscript has been deemed suitable for publication in PLOS ONE. Congratulations! Your manuscript is now with our production department. 

Kind regards, 

on behalf of

Dr. Khin Thet Wai 

Academic Editor

PLOS ONE